# MoQ: Mixture-of-format Activation Quantization for Communication-efficient AI Inference System

**Haonan Wang**[*]
USC Information Sciences Institute
University of Southern California
Marina Del Rey, CA, USA
haonanwa@usc.edu

**Zeli Liu**[*]
Department of Electrical and Computer Engineering
University of Southern California
Los Angeles, CA, USA
zeliliu@usc.edu

**Chao Fang**
School of Electronic Science and Engineering
Nanjing University
Nanjing, China
fantasysee@smail.nju.edu.cn

**John Paul Walters**
USC Information Sciences Institute
University of Southern California
Arlington, VA, USA
jwalters@isi.edu

**Stephen Crago**
USC Information Sciences Institute
University of Southern California
Arlington, VA, USA
crago@isi.edu

## Abstract

In the era of AI, a drastic expansion of model size is witnessed, which poses challenge to the resource efficiency of AI system. To benefit communication-sensitive applications on distributed edge AI inference systems, quantization has been widely applied as a promising technique to compress the activation. Existing mainstream works exploit either integer (INT) or float-point (FP) format of quantization for the entire model. However, they overlook the possibility of jointly leveraging mixture of formats and exploring their strength accordingly tailored to different models and layers. In this work, we first comprehensively analyze the feature of different formats of quantization, including both INT and FP format, with or without clipping method, and draw some insights about the advantages of different formats under different circumstances. Then, we propose a lightweight calibration-based Mixture of format Quantization (MoQ) strategy that enables a communication-efficient AI serving system to automatically adapt to different models with different activation distributions, and achieve minimal accuracy loss using optimal format of quantization for different layers. To quantitatively evaluate the capability of locating the optimal format for different layers of our MoQ format selection strategy, a criteria termed Optimum Hit Rate (OHR) is defined. Experimental results show that by leveraging our proposed MoQ method, an AI inference system can achieve significant improvement of OHR over any static format quantization and intermediate measurement-based strategies.

**Keywords:** Transformer, CNN, communication efficiency, quantization, mixture of format

---

[*]contribute equally

38th Second Workshop on Machine Learning with New Compute Paradigms at NeurIPS 2024(MLNCP 2024).

# 1    Introduction

Large-scale AI models, especially represented by Transformer models [Vaswani, 2017], have emerged as a dominant technique for numerous tasks, such as natural language processing (NLP) [Touvron et al., 2023] and computer vision (CV) [Dosovitskiy, 2020, Ma et al., 2024, Ye et al., 2024, Wang et al., 2020]. However, great performance comes at the price of an explosion in model resource requirements. This poses severe challenges to their deployment at the edge, where computation and memory resources are restrictively constrained. To address this issue, integer quantization (INT) [Frantar et al., 2022, Yao et al., 2022, Tian et al., 2023] has been used to compress the weight data, allowing inference with reduced memory. However, with the rapid development of neural networks, new model structures, such as Transformer, and new feature learning mechanisms, like attention layer [Vaswani, 2017] introduce new data distributions, making the quantization for AI models more challenging. Many works [Wei et al., 2023, Wang et al., 2023] pinpoint the difficulty of high-precision INT quantization for Transformer models, because of the widespread existence of outliers in activations. This large variance in distribution results in a wider dynamic quantization range, forcing the INT quantization to waste its bits to represent a small proportion but immense outliers, leading to a larger quantization error. Thus, the clipping method [Banner et al., 2019] is introduced to tackle this problem by confining an optimal quantization range outside of which the outliers will be clamped by the boundary value. Alternatively, float-point (FP) format [Rouhani et al., 2023, Wu et al., 2023] also reports better quantization precision than INT in handling outliers due to its intrinsic wide dynamic range and non-uniform representation. Most of the aforementioned works only focus on applying a specific format using one single bitwidth for the entire model, but works [Jin et al., 2020, Tang et al., 2022b, Bulat and Tzimiropoulos, 2021, Tang et al., 2022a] point out that the distribution of data varies drastically across different models and layers; thus they explore different bitwidth tailored to different layers. Unfortunately, even though these works have investigated the dimension of mixed bit precision, they still do not promise an optimal quantization accuracy, since they overlook the other two important dimensions of quantization, which are the combination of formats (i.e., INT, and FP formats with different bit allocations), and whether to apply clipping.

In this work, we first conduct a comprehensive study of full spectrum of quantization formats for communication compression, including arbitrary bitwidth of INT and FP quantization, as well as all bit allocations of FP formats. We also investigate the impact of the clipping method for both INT and FP quantization. Furthermore, We unveil the strong correlation between the final output layer's quantization MSE and the model accuracy, enabling accurate prediction of optimal quantization format before system runtime. Based on this finding, we propose a Mixture-of-format activation Quantization (MoQ) method, which employs a lightweight calibration procedure that locates and employs the optimal quantization format for different layers and models, to achieve the best accuracy for communication-efficient AI inference systems. To evaluate the performance of each format, we define a criterion that represents how accurately a strategy can locate the optimal format, termed optimum hit rate (OHR). Experimental results illustrate that the proposed MoQ significantly outperforms all other strategies w.r.t. OHR.

# 2    Background

***INT and FP Quantization.*** In neural networks, data is typically trained and stored using full-precision numbers, providing high precision but also incurring significant memory and communication overhead. Quantization addresses this by reducing the bitwidth to represent each value. In general, quantization can be categorized into INT and FP quantization, which quantizes the original full-precision values $x$ to lower-bitwidth INT or FP numbers $x_q$, respectively.

In this work, we focus on investigating the features of different formats instead of the quantization paradigm itself. For INT format, we adopt a simple quantization paradigm:

$$x_q = \text{Round}\left(\frac{x}{s}\right) \times s; \quad s = \frac{\max(|x|)}{2^{b-1} - 1} \tag{1}$$

where $s$ is the scaling factor determined by the maximum absolute range, and $b$ is the bit budget.

For FP format, we adopt a simple but effective FP format from AdaptivFloat [Tambe et al., 2019], which dynamically maximizes its available dynamic range in a layer-wise fashion, ensuring an accurate encoding of neural network parameters.

***Clipping Method.*** The presence of outliers in neural networks, especially in activation [Wei et al., 2023], can significantly impact the effectiveness of the quantization performance. These outliers can disproportionately affect the quantization range when the scaling factor is determined by the maximum absolute value, leading to significant quantization errors. To address this issue, various clipping methods [Choi et al., 2018, Banner et al., 2019] are proposed to enhance INT quantization. The clipping process involves setting a threshold value $c$, beyond which data points are clipped to the threshold:

$$x_{\text{clipped}} = \min(\max(x, -c), c) \tag{2}$$

where $x$ is the original data point, and $x_{\text{clipped}}$ is the clipped data. This formula ensures that all data points are restricted within the range $[-c, c]$. Then the outlier-suppressed $x_{\text{clipped}}$ can be further processed by quantization methods. We adopt the ACIQ [Banner et al., 2019] method in our work, since it is a post-training method without the need of training, and it provides an analytical optimal clipping threshold that minimizes the quantization error.

## 3 MoQ: Mixture of Format Activation Quantization

### 3.1 Motivation of layer-wise mixture-of-format quantization

For a communication-efficient distributed AI inference system, a mixture-of-format quantization is feasible but often overlooked. This scenario does not require specific hardware support for all possible formats; it only necessitates a bit-packing/unpacking implementation at the communication interface. When the system allocates a bit budget to a certain layer due to bandwidth constraints, it can benefit from flexible communication compression while achieving an improved accuracy with the optimal quantization formats. In light of this consideration, we propose **Mixture-of-format Quantization (MoQ)**. The main idea of MoQ is to apply different quantization formats tailored to different layers' specific distributions. This approach jointly leverages the strengths of each format to achieve optimal compression and minimal quantization error. Our MoQ method covers the full spectrum of format combinations, including arbitrary bitwidth of both INT and FP quantization, as well as an arbitrary bit-allocation of FP formats. We also study the impact of the clipping method along with both INT and FP quantization. We comprehensively investigate the feature of each combination of the mixed format, and design a calibration-based mechanism to determine the optimal formats.

### 3.2 Light-weight Calibration-based MoQ strategy

Since different layers and models might possess significantly different distributions, quantization performance varies for different formats. It will introduce a large amount of engineering work to manually select the optimal format for every model or layer type that will be executed on an AI system. Hence, an intelligent strategy that can spontaneously locate the optimal format is highly required but overlooked by existing works. It calls for an approach that can precisely predict the impact on the model accuracy brought by quantization.

To explore whether it is feasible to predict the quantization impact, a natural idea is to probe the correlation between the error level introduced by quantization at the intermediate quantized layer and final output layer and the final accuracy. The three columns of Fig. 1 show applying 4-, 6-, and 8-bit INT quantization, respectively, to each layer of the ViT-Base model [Dosovitskiy, 2020], a popular Visual Transformer model. In each sub-figure, the x-axis denotes the layer ID at which the activation is quantized. We measure the layer-wise error of mean square error (MSE) to evaluate how quantization affects the model performance. The sub-figures in row (a) represent the MSE of the intermediate full-precision activation and its quantized version, while sub-figures in row (b) illustrate the MSE of the original output tensor of the entire network and its counterpart when quantization is performed at that intermediate layer. Row (c) denotes the final model accuracy evaluated on the validation set of ImageNet [Deng et al., 2009].

By comparing rows (a) and (c), unfortunately, there is no obvious correlation between the intermediate MSE and the final output, which indicates any attempt based on immediate measurement at

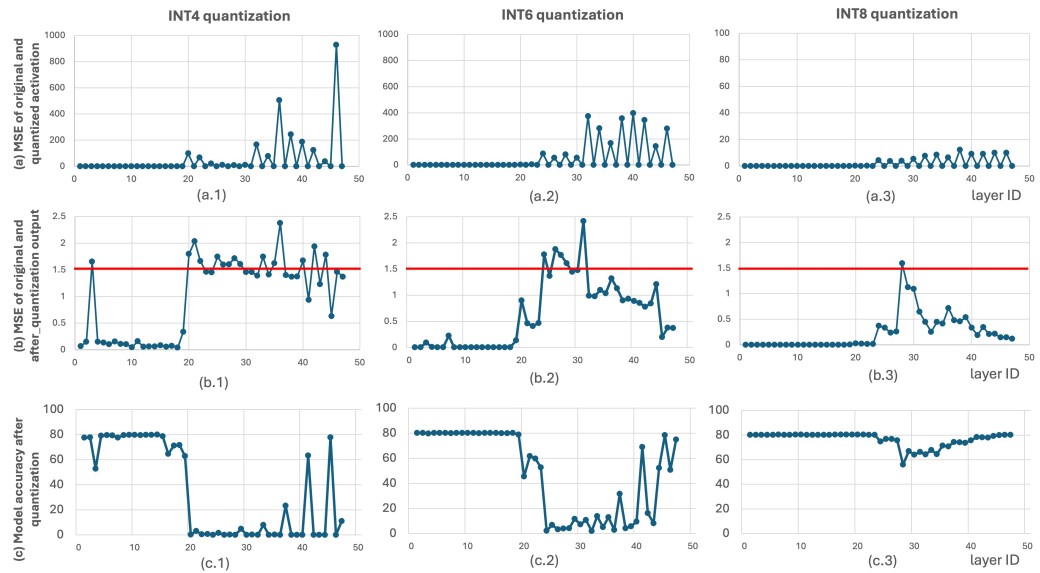

Figure 1: Correlation between MSE of intermediate quantized/final output layer and model accuracy.

runtime is not feasible. We also test other error measurement criteria, like cosine distance [Wu et al., 2020], and also observe a similar poor correlation as MSE.

To effectively evaluate the optimal format, we further investigate the correlation between the final output layer's MSE and the model accuracy by comparing rows (b) and (c). Inspiringly, we see a strong correlation. We find an empirical threshold of MSE with the value of 1.5, marked by the red line in the figures. For layers with MSE larger than this threshold, it will result in very poor accuracy that is close to zero. Thus, if we truncate the data points that are above this threshold in sub-figures of row (b), and then flip the curve upside down, we will observe that it shares a similar shape to the accuracy curve in row (c), which indicates a relatively strong correlation between the MSE of the final layer's output and the model accuracy. This result implies that it is possible to predict the final model performance by evaluating the MSE of the output layer. It is reasonable because the quantization error introduced to the output tensor will directly reflect the distortion of final classification vector. But if we measure the MSE of the intermediate quantized tensor, the errors will be propagated through the remaining layers, which results in an indirect and inaccurate relation to the classification accuracy. However, evaluating the distortion in the output layer requires the execution of the original model and quantized models covering all quantization formats. It is not practical to run this measurement during the system runtime since it will bring huge computation overhead multiple times, canceling out the effort of inference acceleration via communication compression.

In order to balance the format prediction performance and overhead, we propose a lightweight calibration-based MoQ method. Evaluating the prediction on a small bunch of samples of the dataset, termed calibration set, can yield the choice of quantization format that achieves the optimal or near-optimal model accuracy. Here, we make two fundamental assumptions to justify the rationality of MoQ: **(a.)** The distribution of the input to a system is independent and identically distributed (i.i.d), it inspires that we can leverage a calibration dataset to probe the error level on the final output brought by quantization at a certain layer; **(b.)** The distribution of activation data of different layer change insignificantly with different inputs, so the optimal choice of quantization stays consistent for different input. Based on these assumptions, the optimal quantization format can be determined by designing a calibration phase in the AI inference system at which all quantization formats are evaluated on a lightweight calibration dataset. To further explore the tradeoff between model accuracy and system performance, we investigate the relation between prediction accuracy and the size of the calibration dataset. Moreover, since we now have a dataset to be evaluated, instead of measuring the MSE of the output layer, which is an indirect proxy to the target metric, i.e., model accuracy, we determine the optimal format based on the evaluation accuracy on the calibration set during the calibration phase. This measurement directly contributes towards the target metric;

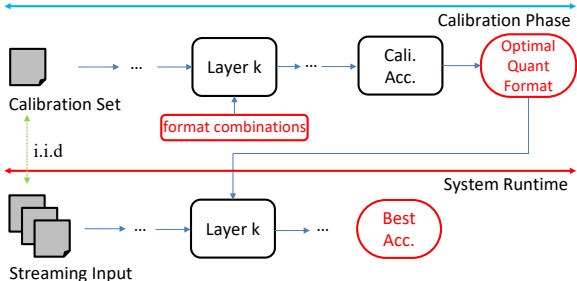

Figure 2: The workflow of MoQ. A constant-running AI inference system leverages a calibration phase to determine the optimal quantization format and deploys it during the runtime.

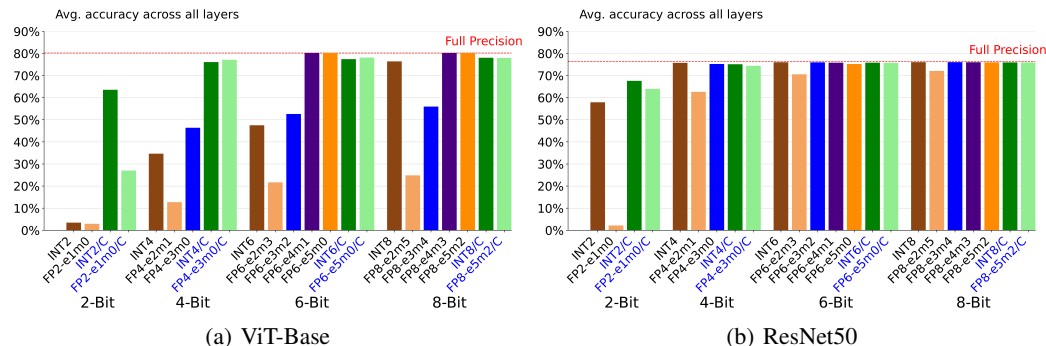

Figure 3: Average accuracy vs. full-spectrum quantization formats on Transformer and CNN model.

thus, it is more consistent and accurate while even introducing less computing overhead compared to computing MSE of the output tensors. The overall workflow of the MoQ is shown as Fig. 2. For a constant-running AI inference system, it can enter the calibration phase periodically to update the optimal formats. Experimental results show that our MoQ method can achieve good prediction with only a small calibration size, which brings very little overhead. More details are exposed in Table 4.

## 4 Experimental Evaluation

### 4.1 Experiment Testbed and Design

Our target application scenario for the proposed MoQ is a communication-efficient distributed AI inference system. It only applies quantization to activation data for communication compression using the optimal quantization format for each layer. By applying MoQ, the system can benefit from significant communication compression, while also achieving the optimal accuracy resulting from the better representative capability of the optimal format combination. Thus, we evaluate our MoQ methods on a state-of-the-art distributed edge AI serving system, called QuantPipe [Wang et al., 2023], which provides a pipelined testbed that simulates the communication constraints to test different quantization strategies. The system is operated on Linux with kernel version 3.10.0. We implement the MoQ encode/decode module in the QuantPipe system based on Python 3.8 and the PyTorch 1.12 framework.

We explore multiple quantization strategies to compress the communication with various bit budgets, which are determined by the AI serving system's bandwidth constraints. The experiments consider both INT quantization and FP quantization and cover bitwidth ranging from 2 to 8 bits. For FP quantization, we also investigate the most reasonable exponent and mantissa allocations since different bit allocations can result in considerable accuracy deviation. Besides, we further add another dimension of exploration regarding whether to use the clipping method. To cover the mainstream model architectures, we implement MoQ on two model categories: (a) Visual Transformer (ViT) for transformer architecture and (b) ResNet for CNN architecture.

## 4.2 Static Format Analysis

We first evaluate the capability of different quantization formats to handle various distributions in Fig. 3. The y-axis denotes the average model accuracy across all layers, and the x-axis represents the quantization format. The baseline accuracy of the full-precision model is marked by red dashed lines. We denote $x$-bit INT quantization by INT$\{x\}$, and $x$-bit FP format with $a$-bit exponent and $b$-bit mantissa by FP$\{x\}$-$e\{a\}m\{b\}$. If the clipping method is applied, a suffix $/C$ will be added.

***INT vs. FP quantization.*** For the ViT model, as shown in Fig. 3 (a), when the bit budget is adequate (bitwidth $\geq$ 4), FP formats with exponential bitwidth larger than 3 clearly outperform their INT counterpart. This is because the FP with enough exponential bits provides a wider dynamic range to represent extremely large outliers accurately. On the contrary, INT format and FP with limited exponential bits ($\leq$ 2) cannot handle the outliers without harming the representative precision of small values that account for a large proportion of the data, leading to degraded model accuracy. But this superiority of FP format is not obvious for the CNN model as reported in Fig. 3 (b), because the presence of such large-magnitude outliers in transformer models is less pronounced in CNNs [Wei et al., 2024]. Moreover, when the bit budget is limited (i.e., 2 bits), INT outperforms FP on both ViT and ResNet. It is because FP2-e1m0 requires one bit for the sign bit, so there are no sufficient bit budgets for the exponent to represent the magnitude and for mantissa to maintain precision. The performance gap of INT2 between ViT and ResNet results from their drastic difference in the data range. In general, FP formats with sufficient bit budget are more suitable for models with large distribution variance, like the Transformer model, while INT can better model the low-variance data under limited bitwidth.

***Bit allocation of FP format.*** According to Fig. 3, we can empirically conclude that the exponential part is more important than mantissa for activation quantization. It shows the model accuracy will increase as the exponential bitwidth increases from zero bit, and then when the exponential bits go beyond a certain value (4 bits for ViT models and 3 bits for ResNet models), the model accuracy will saturate with only trivial difference for different mantissa bitwidth. This finding inspires the idea that for FP format quantization, it is important to select a suitable lower bound for exponential bits, considering the target model's data range.

***Effect of Clipping.*** Clipping is employed to clamp outliers, allowing the quantization process to focus on the central data distribution. Formats with clipping are denoted with the suffix $/C$ in our experiments. We report their performance in Fig. 3 (a) and (b). It can be observed that clipping is particularly beneficial when the bit budget is extremely limited ($\leq$ 4 for ViT and $\leq$ 2 for ResNet). By eliminating outliers, clipping can significantly reduce quantization error under a limited bit budget. However, when the bit budget is sufficient, clipping instead tends to harm the accuracy slightly. It is because the error introduced by clipping the outliers exceeds its benefit to quantization. This result indicates that clipping is useful for limited bit budgets but can harm the accuracy when bitwidth is adequate.

## 4.3 Lightweight calibration-based MoQ

The above rules are helpful in guiding the selection of a good format for quick deployment purposes. However, in real applications with the requirement of optimal model accuracy, it is not convincing to adopt a heuristic format without guaranteed optimum, and it is not feasible to run complete evaluations at runtime to confirm whether the selected format is optimal. Thus, we propose the lightweight calibration-based MoQ strategy to yield the optimal quantization format choices automatically.

As shown in Table 1, to showcase the superiority of the MoQ strategy, we compare it with monolithic INT and FP quantization, with and without ACIQ clipping. We also evaluate the prediction methods based on intermediate error measurement using MSE and Cosine [Wu et al., 2020] criteria. To quantitatively compare the optimum prediction performance of each approach, we define an index termed Optimum Hit Rate (OHR). It is calculated by the ratio of the number of layers at which the method successfully predicts its optimal format and the total number of layers of that model. This index shows how well a format-selection strategy can predict the optimal format across different distribution layers. The referenced optimal format is predetermined by evaluation of all formats on the entire ImageNet validation set. Moreover, a format is considered as optimal if its accuracy is within a tolerance range below the accuracy of the optimal format. We report the OHR of all formats

Table 1: Optimum Hit Rate (OHR) for different format strategy

| | | ViT-Base | | | | | ResNet50 | | | | |
|---|---|---|---|---|---|---|---|---|---|---|---|
| Format Strategy | | 2 bit | 4 bit | 6 bit | 8 bit | Avg. | 2 bit | 4 bit | 6 bit | 8 bit | Avg. |
| Static INT | | 2% | 19% | 38% | 57% | 29% | 26% | 100% | 100% | 100% | 81% |
| Static FP[†] | | 0% | 14% | 48% | 63% | 31% | 0% | 33% | 60% | 72% | 41% |
| Static INT/C | | 97% | 59% | 76% | 78% | 77% | 61% | 69% | 100% | 100% | 82% |
| Static FP/C[†] | | 4% | 43% | 53% | 59% | 40% | 42% | 42% | 79% | 82% | 61% |
| MSE | | 2% | 38% | 55% | 68% | 41% | 50% | 71% | 100% | 100% | 80% |
| Cosine [Wu et al., 2020] | | 4% | 36% | 55% | 65% | 40% | 44% | 36% | 100% | 100% | 70% |
| Calibration-based MoQ | 64[*] | 96% | 78% | 90% | 91% | 89% | 71% | 74% | 92% | 94% | 83% |
| | 192[*] | 98% | 84% | 93% | 95% | 93% | 81% | 79% | 94% | 96% | 88% |
| | 320[*] | 98% | 87% | 95% | 96% | 94% | 86% | 82% | 95% | 97% | 90% |
| | 640[*] | **99%** | **90%** | **97%** | **97%** | **96%** | **91%** | 86% | 97% | 98% | **93%** |

[†] Average OHR over all bit allocations; [*] Calibration dataset size

using a tolerance range of 1% in Table 1, which is a reasonable range without harming the users' experience.

Although the calibration-based MoQ evaluation is executed at a separate calibration phase, which only brings about a small overhead to the overall running time when the AI system is up and running for consistent services, we also investigate the impact of the size of the calibration dataset on the prediction OHR. The results show that a lightweight calibration set is enough to yield an accurate prediction of format, considering a size of 640 samples only accounts for around 1% of the entire ImageNet validation dataset.

As shown in Table 1, experimental results indicate that our proposed calibration-based MoQ method shows great performance. Especially for transformer models, we can see that our proposed MoQ methods achieve an average OHR of 96%, which significantly exceeds both static format and intermediate measurement strategies. We investigate the reason and find out that when compared to CNN models, the data distribution of transformer architecture varies drastically across layers due to its attention mechanism and GeLU activation, making it hard for one single format to operate well for all layers. The intermediate measurement prediction methods based on MSE and Cosine do not show clear improvement over static format, because there is no strong correlation between the intermediate error level and the final model performance, as we illustrated in Fig. 1. For CNN models, the proposed MoQ method also shows improved average OHR over all other methods. Note that, for some bit budget cases, e.g., 6 and 8 bits, our method does not achieve the highest OHR (but still very close to 100%). This is because the activation distribution of CNN architectures is consistent across layers and does not possess large-magnitude outliers like transformer models. So it is less difficult for a format with a sufficient bit budget (6/8 bit) to be considered optimal, given that the definition of optimum is its accuracy is within a tolerance range of 1% in our experiments.

## 5  Conclusion

In this paper, we propose the Mixture-of-format activation Quantization (MoQ) method, aimed at improving communication efficiency and accuracy in the distributed AI inference system. By locating the optimal format from full-spectrum formats based on calibration-based evaluation, our MoQ method can effectively handle various distributions of different layers of different models, improving the model accuracy while maintaining its communication efficiency.

## Acknowledgment

This work was partially supported by the Department of the Navy, Office of Naval Research with grant #N00014-20-1-2143, and by an ISI Exploratory Research Award. Any opinions, findings, conclusions, or recommendations expressed in this material are those of the author(s) and do not necessarily reflect the views of the funding agencies.

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
