# OpenReview forum: "MoQ: Mixture-of-format Activation Quantization for Communication-efficient AI Inference System"
_NeurIPS.cc/2024/Workshop/MLNCP — MLNCP Poster_

### Official Review · Reviewer_b1mY · 2024-10-03

**Rating:** 7
**Confidence:** 5

**Review:**

Paper content summary: The paper thinks outside the box of fixed/mixed precision and proposes to combine recent works of new data formats and outlier clipping methods into the search grid for optimizing accuracy, based on the observation that layers in a model have different distributions.

Reasons to Accept:

- novelty:
    - novel idea of a mixture of formats instead of a simple mixture of precision. new results on mixing with FP quantization and clipping methods.
- insight:
    - solid and insightful analysis of the strength of different formats and when to use them in Section 4.2
    - solid analysis of the correlation between loss on intermediate activations, final output, and accuracy, in section 3.2
- impact:
    - targets a clear and important application scenario: communication-constrained edge system. The studied models include both the older and smaller CNN/ResNet and newer and larger ViT, which have different distributions. They cover most of the edge visual modeling applications.
    - practical for deployment as the calibration is super lightweight (i.e. no backpropagation, just grid search), though I’m curious how feasible it is for hardware and system to support mixture of formats deployment.
- method:
    - reasonable and well-documented

Main Concern:

- it is not clear how ***significant*** the improvement is by using MoQ rather than static format. the effectiveness of MoQ compared to static format is evaluated by OHR metric the authors came up with. But OHR is a discrete and indirect measure, the result of which varies highly with the selected loss tolerance. A more direct and convincing measure would be to show if accuracy on evaluation datasets (e.g. imagenet) is improved using MoQ instead of static format.
    - I also had doubts due to Table 1. showing MoQ can sometimes has lower OHR than static formats. Why would MoQ be worse in any case? It searches for all possible formats which include the static format choice, and picks the optimal based on the final output MSE. If the MSE is a good proxy, theoretically MoQ will always be better than the static format. Is the gap due to a lower correlation between final output MSE and accuracy in these cases?

Score: 7 and open to raising it after the main concern is addressed

Potential future work:

- Currently, in your work, the precision is set by the user and then the optimal format is found by the system. But different layers may have different optimal precision as well. Ideally allowing both mixture of formats and mixture of precision may yield a better efficiency and quality trade-off.

---

### Official Review · Reviewer_pjrn · 2024-10-07
**An interesting and useful result, although I am unsure if this has been done before.**

**Rating:** 7
**Confidence:** 3

**Review:**

This paper is well motivated by the problem of quantization in low-precision DNNs, in light of data movement or hardware constraints.  They did a good job showing that quantization error can be accurately correlated to the output activation MSE, and does not correlate well to weight MSE, and then come up with a clever technique to use a "calibration dataset" in order to estimate the former, a lightweight operation that allows quantization to be done without additional training.

The details of the calibration method were a little bit lacking.  I feel that it might be something related to linearization of the model, but am not sure.  How do you figure out how the weight errors affect the calibration outputs, without running everything through the full neural network (which I thought was the point of this paper).  There is also no public codebase for this paper.

In references, please never, ever, ever refer to Communications Physics as “Nature Communications Physics”

Evaluation: interesting result but not groundbreaking.  The field of quantization is very large so it can be difficult to determine whether similar things have been done before.  50-60th percentile.

---

### Decision · Program_Chairs · 2024-10-10

Accept (Poster)